# Use of Omics Data in Fracture Prediction; a Scoping and Systematic Review in Horses and Humans

**DOI:** 10.3390/ani11040959

**Published:** 2021-03-30

**Authors:** Seungmee Lee, Melissa E. Baker, Michael Clinton, Sarah E. Taylor

**Affiliations:** 1The Dick Vet Equine Hospital, The Roslin Institute, Easter Bush, Roslin, Midlothian EH25 9RG, UK; melissa.baker@ed.ac.uk (M.E.B.); sarah.e.taylor@ed.ac.uk (S.E.T.); 2The RICE Group, Division of Gene Function and Development, The Roslin Institute, Easter Bush, Roslin, Midlothian EH25 9RG, UK; michael.clinton@roslin.ed.ac.uk

**Keywords:** microRNA, repetitive load injury, bone, biomarker, RNA-seq, SNP

## Abstract

**Simple Summary:**

Despite many recent advances in imaging and epidemiological data analysis, musculoskeletal injuries continue to be a welfare issue in racehorses. Omics studies describe the study of protein, genetic material (both DNA and RNA, including microRNAs—small non-coding ribonucleic acids) and metabolites that may provide insights into the pathophysiology of disease or opportunities to monitor response to treatment when measured in bodily fluids. As these fields of study are scientifically complex and highly specialised, it is timely to perform a review of the current literature to allow for the design of robust studies that allow for repeatable work. Systematic reviews have been introduced into the medical literature and are a methodological way of searching for relevant papers followed by critical review of the content and a detection of biases. The objectives of the current systematic review were to identify and critically appraise the literature pertaining to microRNA (miRNA) and their target genes that are correlated with stress fractures in racehorses and humans. The object was to define a panel of miRNAs and their target genes as potential biomarkers in either horses or human subjects. The online scientific databases were searched and a reviewed was performed according to preferred reporting items for systematic reviews and meta-analyses (PRISMA) guidelines. MicroRNA profiling studies in horses continue to emerge, but as of yet, no miRNA profile can reliably predict the occurrence of fractures. It is very important that future studies are well designed to mitigate the effects of variation in sample size, exercise and normalisation methods.

**Abstract:**

Despite many recent advances in imaging and epidemiological data analysis, musculoskeletal injuries continue to be a welfare issue in racehorses. Peptide biomarker studies have failed to consistently predict bone injury. Molecular profiling studies provide an opportunity to study equine musculoskeletal disease. A systematic review of the literature was performed using preferred reporting items for systematic reviews and meta-analyses protocols (PRISMA-P) guidelines to assess the use of miRNA profiling studies in equine and human musculoskeletal injuries. Data were extracted from 40 papers between 2008 and 2020. Three miRNA studies profiling equine musculoskeletal disease were identified, none of which related to equine stress fractures. Eleven papers studied miRNA profiles in osteoporotic human patients with fractures, but differentially expressed miRNAs were not consistent between studies. MicroRNA target prediction programmes also produced conflicting results between studies. Exercise affected miRNA profiles in both horse and human studies (e.g., miR-21 was upregulated by endurance exercise and miR-125b was downregulated by exercise). MicroRNA profiling studies in horses continue to emerge, but as yet, no miRNA profile can reliably predict the occurrence of fractures. It is very important that future studies are well designed to mitigate the effects of variation in sample size, exercise and normalisation methods.

## 1. Introduction

Musculoskeletal injuries in racehorses are common and associated with potential welfare issues. In a recent meta-analysis, the pooled incidence of catastrophic musculoskeletal injury was 1.17 per 1000 race starts [1]. At least one fatal outcome was reported at 21.5% of National Hunt events, with bone injuries being the most common [2]. The majority of deaths reported in this study resulted from injuries to bone (77.8%), and these predominantly involved the distal limb, with the third metacarpal or metatarsal as the most commonly affected structure [2]. Similarly, in flat racing, the majority (77%) of fatalities of Thoroughbreds in Great Britain were associated with a bone injury [3]. Early detection of incomplete fractures can be difficult, and whilst imaging represents the frontline approach to the screening of racehorses, there is currently no consensus about the specific interpretation of various imaging modalities including radiography, nuclear scintigraphy, computed tomography, magnetic resonance imaging and positron emission tomography [4].

Stress fractures arise because of an imbalance between damage accumulation and targeted repair at predictable sites of repetitive bone injury. Despite the progress in advanced imaging, micro-CT [5] and MRI [6] have failed to predict why certain levels of bone densification and microdamage that lack targeted repair go on to propagate to complete fracture in some individuals, yet the same levels are tolerated by others [7].

Omics technologies, genomics, transcriptomics, proteomics and metabolomics represent the study of the molecules, DNA, RNA, protein and metabolites, respectively. Genetic association studies aim to identify if a certain gene (DNA) controls the phenotype of a particular disease. Transcriptomic studies use next generation sequencing to identify if messenger RNA (mRNA) influences a particular disease. MicroRNAs (miRNAs) are small non-coding RNA molecules that are able to influence post-transcriptional gene expression. MicroRNAs have been proposed as potential diagnostic and prognostic biomarkers [8,9] in many diseases, and numerous miRNAs have been shown to be dysregulated in various different types of cancers as oncogenes or tumour suppressor genes [10]. In addition to their potential as biomarkers, miRNAs are being evaluated for their use as therapeutics in some cancers and in treating hepatitis C. Several miRNA-targeted therapeutics in humans have reached clinical trials, including a tumour suppressor miRNA-34a mimic in Phase I clinical trials [11,12] and miR-122 anti-miRNA in Phase II clinical trials for treating hepatitis C infection [13].

As circulating miRNAs have been identified as potential biomarkers of fractures in osteoporosis [14,15,16,17,18] and are beginning to be evaluated in equine orthopaedic disease [19,20,21,22,23,24], the analysis of circulating miRNAs represents an exciting opportunity to study musculoskeletal health, yet currently, there are no standardised pre-analytical, analytical and post-analytical guidelines to allow for comparison between studies. It is therefore timely to conduct a review of the literature. Whilst the aetiopathogenesis of equine stress fractures secondary to repetitive loading is clearly different to that of osteoporotic stress fractures, it was hoped that parallels in study design and analysis could be drawn. The authors chose to consider both equines and humans, as both species sustain stress fractures [25] and their bones have been shown to heal in situ without removal of the damaged domain [26].

Systematic reviews use explicit methods to identify, select, appraise and synthesise results from similar but separate studies to identify high-quality evidence and highlight gaps in the current research. Scoping reviews are essential where there is a large and diverse evidence base, to provide a broad overview of the current evidence and to identify areas suitable for more detailed evaluation in a systematic review. Currently there are only two narrative reviews of miRNA relating to veterinary orthopaedics [27,28], and no published systematic reviews or scoping reviews pertaining to stress fractures. There are a range of different frameworks that have been developed to optimise the process of systematic reviews. Preferred reporting items for systematic reviews and meta-analyses (PRISMA) is widely accepted as the methodological framework for systematic reviews and is recommended by many journals [29,30]. PRISMA provides an evidence-based minimum set of items that should be evaluated and reported, and their resources include a standardised checklist and flow diagram.

It has been recommended that miRNA–RNA interactions should be validated in appropriate biological systems [31]. Analysis of miRNA–target interactions has been performed with respect to bone using the following approaches: (1) target prediction programmes followed by mRNA detection in bone tissue [17]; (2) correlations of differentially expressed miRNAs with bone turnover markers [32]; (3) transfection of cells with miRNA and assessment of in vitro osteogenic potential, such as bone alkaline phosphatase assays and Alizarin red staining [33]; (4) treatment of mice with an agomir of the miRNA of interest and an evaluation of bone density [18].

The objectives of this review are to (1) make recommendations regarding blood sampling for miRNA studies with respect to the timing of feeding, exercise and haemolysis; (2) summarise the relevant omics data pertaining to stress fractures in horses and humans; (3) define and compare the panels of miRNA biomarkers and their miRNA targets identified in studies of osteoporotic fractures in humans to provide methodological insights for equine research.

## 2. Materials and Methods

### 2.1. Protocol and Registration

The review followed the methodology described in the Cochrane handbook for systematic reviews and the PRISMA statement for reporting systematic reviews and meta-analyses [29,30]. This protocol was modified based on previously published systematic review papers [34,35]. The preferred reporting items for systematic reviews and meta-analyses protocols (PRISMA-P) checklist is presented in Appendix A. The intervention for this scoping and systematic review was stress fractures in horses and the outcome under consideration was the concentration of serum/plasma miRNA in horses. As no literature was identified measuring miRNA in horses with stress fractures, the intervention was broadened to include stress fractures in humans. Stress fractures in horses occur frequently in young athletes during racing or training. Similar injuries also occur in humans, especially elite athletes or military recruits. Since there are only limited miRNA studies in horses, core findings in systematic human studies could also be valuable for a better understanding of the horse studies. Consequently, the outcomes included plasma/serum miRNA concentration in horses and humans. Widening the search for this systematic review was considered appropriate, as stress fractures that occur in young equines also occur in elite athletes and military recruits [25]. When the literature was searched for stress fractures in humans and horses, again there was a paucity of studies. As the objectives of this review included making recommendations regarding blood sampling for miRNA studies with respect to timing of feeding, exercise and haemolysis, the search inclusion criteria were further broadened to include omics studies in normal exercising horses and those with musculoskeletal disease.

### 2.2. Synthesis of Outcomes for the Systematic Review

The three primary outcomes were to identify (1a) changes microRNAs/peptide analysis/gene expression related to musculoskeletal injuries in horses, (1b) microRNAs and their targets in response to exercise and mechanical loading in horses and humans and (1c) genetic association studies related to stress fractures in horses and young adults (e.g., athletes/military recruits) equivalent to equine stress fractures. The secondary outcomes were to analyse miRNA and their targets related to osteoporotic fragility fractures in humans.

### 2.3. Eligibility Criteria

We included randomised controlled trials (RCTs), and cohort, case-control and cross-sectional studies. We excluded case series, case reports, narrative reviews and textbook chapters. Study definition and categorisation were based on the Joanna Briggs Institute (JBI) reviewer’s manual and methods for the development of the National Institute for Health and Care Excellence (NICE) public health guidance. A randomised controlled trial (RCT) is a study where participants are randomly allocated to receive either the intervention or the control. A quasi-experimental study or non-randomised controlled trial is a study where participants are allocated to receive either the intervention or the control, but the allocation is not randomised, an approach often called a controlled before-and-after or a time-series study. A cohort study is an observational study in which a group of people or animals (cohort) are observed over time in order to see who develops the outcome longitudinally. A case-control study is a study where the investigator selects people or animals who have an outcome of interest (e.g., disease) and others who do not (controls), and then collects data to determine previous exposure to possible causes. A cross-sectional study is an observational study in which the source population is examined to see what proportion has the outcome of interest, or has been exposed to a risk factor of interest, or both [36,37]. A study was included if the full text could be obtained from any of the University of Edinburgh libraries or e-libraries, through University of Edinburgh journal subscriptions, or from free online Open Access sources.

### 2.4. Information Sources

#### 2.4.1. Databases

The following databases were searched for this study:Medline In-Process and Non-Indexed Citations and Ovid MEDLINE: 1946–present;WEB of Science (Core Collection: Citation Indexes): 1950–present;SCOPUS: 1966–present.

#### 2.4.2. Search Terms

For the horse studies, the following search terms were utilised for each of the respective databases:

Medline:

horse* OR equine* OR equus* OR racehorse* OR Thoroughbred* AND Fracture* OR fractures* OR catastrophic* OR stress fracture* OR injury* OR orthopaedic* OR bone* OR musculoskeletal* AND miRNA* OR microRNA* OR small RNA* OR RNA-seq* OR GWAS*.

WEB of Science:

TS = (horse* OR equine*) AND TS = (fracture* OR catastrophic* OR stress fracture* OR injury* OR orthopaedic* OR bone* OR musculoskeletal*) AND TS = (miRNA* OR microRNA* OR small RNA* OR RNA-seq* OR GWAS*).

SCOPUS:

TITLE-ABS-KEY (horse* OR equine* AND fracture* OR catastrophic* OR stress fracture* OR injury* OR orthopaedic* OR bone* OR musculoskeletal* AND miRNA* OR microRNA* OR small RNA* OR RNA-seq* OR GWAS*) AND (LIMIT-TO (LANGUAGE, “English”)).

For the human studies, the following search terms were utilised for each of the respective databases:

Medline:

Human* OR athlete* OR military recruit* AND fracture* OR stress fracture* OR fatigue fracture* AND miRNA* OR microRNA* OR small RNA* OR RNA-seq*.

WEB of Science:

TS = (human*) AND TS = (fracture* OR stress fracture* OR fatigue fracture*) AND TS = (miRNA* OR microRNA* OR small RNA* OR RNA-seq*).

SCOPUS:

TITLE-ABS-KEY (human* OR athlete* OR military recruit* AND stress fracture* OR fatigue fracture* AND miRNA* OR microRNA* OR RNA-seq* OR GWAS*) AND (LIMIT-TO (LANGUAGE, “English”)).

### 2.5. Study Selection

A primary literature search of the databases was conducted using the search terms outlined previously. The results of each search were downloaded into bibliographical software EndNote X9 (Thomson Reuters). Duplicates were deleted within EndNote. Publications were then assessed through three stages: review of titles for suitable publications, review of abstracts against inclusion and exclusion criteria and review of the full publications. All titles within the EndNote library were examined and their abstracts were reviewed. Ambiguous titles were retained for further review at the next stage (review of abstract).

Abstracts from these publications were then independently assessed by reviewers for agreement with the inclusion and exclusion criteria for each of the primary and secondary outcomes (Appendix A). Any publications which were ambiguous were retained and reviewed in the next step (review of the full publication by SL and MB). The full text of the final publication confirmed eligibility for this review and progressed to data collection and quality appraisal (Figure 1).

### 2.6. Data Collection Process/Data Items

The final full publications and methodological features were recorded on the detailed data extraction form (Appendix A) based on a data collection form by Cochrane Effective Practice and Organization of Care (EPOC). This was carried out independently by one author (SL).

### 2.7. Quality Appraisal and Risk of Bias in Individual Studies

Included studies were qualitatively assessed by two independent reviewers based on the JBI critical appraisal checklist. Cumulative evidence of each paper was assessed by two independent reviewers (SL and MB) based on the grading of recommendations, assessment, development and evaluations (GRADE) system [38] and vetGRADE system [39] for human and equine studies, respectively. Quality appraisals were assessed by two investigators (SL and MB). When both investigators agreed to each criterion, “yes” was recorded, otherwise, “unclear” or “no” was recorded. The risk of bias was incorporated alongside the reporting of results within the results section of the article.

## 3. Results

### 3.1. Study Selection

The bibliographic searches performed between 24 April 2020 and 7 May 2020 identified 201 and 153 publications in horse and human studies, respectively, after the removal of duplicates. A review of the titles and abstracts was performed against the inclusion and exclusion criteria. Overall, 51 horse and 40 human studies were selected on the basis of their relevance. A further full text review was carried out, which excluded articles for various reasons according to our inclusion/exclusion criteria (Figure 1).

A total of 21 horse studies and 19 human studies met the defined inclusion/exclusion criteria and were assessed with the JBI critical appraisal tools. The PRISMA 2009 flow diagram of the study selection is reported in Figure 1. Data was summarised in a narrative way because of the wide heterogeneity of the study population, study design and analytical methods, it was not possible to perform meta-analysis of data. Determining of differentially expressed miRNAs and genes was based on the results of statistical analysis where the P value, adjusted P or false discovery rate (FDR) value was less than 0.05, depending on the tests carried out. Only significantly differentially expressed miRNAs and their targets were highlighted in cross-comparison studies.

### 3.2. Study Characteristics

The data extracted from the study characteristics consisted of the year of the study, study design (Appendix A), inclusion and exclusion criteria (Appendix A) sex and sample source (Appendix A), methods (Appendix A) and results (Appendix A). The publications included were published between 2008 and 2020. For primary outcome 1a, where 12 papers were reviewed, the most common study designs were cohort studies (4) followed by case-control studies (3) and cross-sectional studies (3), a quasi-experimental study (1) and a randomised controlled trial (1). For primary outcome 1b, where 11 papers were reviewed, there were 10 quasi-experimental studies and 1 randomised controlled trial. For primary outcome 1c, where six papers were reviewed (five human and one horse) there were six case-control studies. For the secondary outcome, where 11 human studies were reviewed, 8 were cross-sectional studies, 2 were case-control studies and 1 was a cohort study.

The horse study population recruited included horses who visited veterinary hospitals (11 studies), healthy horses (8 studies) and post-mortem material (2 horse studies). The sample size of horse studies varied; seven studies had sample sizes of less than 30, nine studies ranged from 30 to 100 and five larger studies had sample sizes ranging from 111 to 529. The mean or median reported age for horses was most commonly from 2 to 7 years old (13 studies), 2 studies did not provide data, 5 studies included horses older than 7 years of age and 1 study reported a mean age of 2 or less. Information of breed or type of horses involved was reported for all studies, with Thoroughbred being the most common type of breed (nine studies). Horse sex was not reported in 5 studies, was balanced in 15 studies and 1 study included only male horses. There were specific exclusion reasons relating to technical issues or confounding disease in 12 horse studies (Appendix A).

The human patients were referred to hospital (17 human studies) and healthy participants (2 human studies). Human studies also varied in scale; four studies had sample sizes less than 30, seven studies ranged from 30 to 100 and eight studies had samples sizes greater than 100. The mean or median reported age for 8 human studies was relatively young, ranging from 20 to 30 years old, while 11 osteoporotic fracture studies involved older individuals ranging from 40 to 80 years old. Human studies revealed more sex bias due to the study population (e.g., military recruits, sports elites, pre- or postmenopausal osteoporotic patients); 6 studies included only male or majority male subjects, 10 studies had a majority of female subjects and 3 studies were of mixed sex. There were specific exclusion reasons relating to technical issues or confounding disease in 14 human studies (Appendix A).

### 3.3. Quality Appraisal and Risk of Bias

All included studies met at least 60% criteria out of total criterial for each study. For primary outcome 1a there was one horse randomised controlled trial study [21], when assessed using the JBI critical appraisal tool for randomised controlled trial studies (Appendix A), the study met 12/13 criteria with the exception of criterion 4 (participants blind to treatment assignment). Of the four cohort studies for outcome 1a, one study [40] met all criteria, one study [41] met 10/11 criteria, except for criterion 2 (clearly defined measurement of exposure) or criterion 7 (reliable measurement) and two studies [42,43] met 9/11 criteria, except for criterion 6 (free of the outcome at the start of the study) or criterion 10 (strategies to address incomplete follow-up). Of the three case-control studies assessed using the JBI critical appraisal tool for case-control studies (Appendix A) one study [20] met 9/10 criteria, one study [44] met 8/10 criteria, except for criteria 6 (confounding factors) and 7, or criteria 7 and 9. One case-control study [45] met 7/10 criteria, except for criteria 4 (validity of exposure measurement), 6 and 7. Two [46,47] of the three cross-sectional studies met all the criteria of the JBI critical appraisal tool for cross-sectional studies; the other study [48] met 5/8 criteria, the exception being criteria 4 (standard measurement), 5 and 6. The quasi-experimental study met all 10 criteria [19].

For primary outcome 1b, there was one human case-control study, reported by Sansoni et al. (2018) [49] that met 9 out of 13 criteria, the exceptions being criteria 2, 4, 5 and 6 (outcome assessors blind to treatment assignment). Overall, 10 studies were assessed using the JBI critical appraisal tool for quasi-experimental studies (Appendix A). Five studies [50,51,52,53] met all 10 criteria, six studies [54,55,56,57,58,59] met 8/9 criteria, with the exception of criterion 4 (control). For primary outcome 1c, there were six case-control studies [60,61,62,63,64,65] that met 9 out of 10 criteria. For the secondary outcome, eight cross-sectional studies [16,17,32,33,66,67] met all criteria and one study [68] met 7/8 criteria, the exception being criterion 6 (strategy to deal with confounding factors). One cross-sectional study [69] met 6/8 criteria, the exceptions being criteria 5 (define confounding factors) and 6. The two case-control studies [18,70] met 9/10 criteria. The cohort study [71] met 10/11 criteria, the exception being treatment groups were not treated identically (Appendix A).

### 3.4. microRNAs/Peptide Analysis/Gene Expression Related to Musculoskeletal Injuries in Horses

Reviewing miRNAs and their targets to equine musculoskeletal injuries identified 12 publications including 2 miRNA candidate studies, 1 miRNA profiling study, 2 mRNA profiling studies and 7 biomarker studies (horse peptide biomarker studies prior to 2008 were not reviewed) (Appendix A). The three miRNA studies reported on diverse disease processes, tendon injury, laminitis and osteochondrosis, and unsurprisingly, did not report on the same miRNAs. One randomised controlled trial that scored highly in the quality appraisal and risk of bias reported a decrease in miRNA-29a levels in tendon injury, yet it should be remembered that participants were not blinded to treatment assignment. The decrease in miRNA-29a was confirmed using an miR-29a mimic in equine tenocytes, which selectively targeted to COL3A1 encoding type III collagen [21]. Another study reported that levels of miR-23b-3p, miR-145-5p and miR-200b-3p increased in acute laminitis [20]; however, the case-control study of acute laminitis may have been subjected to age biases as the cases were mature horses with laminitis and the controls were immature horses. Targets of these miRNAs were linked to the glutamatergic pathway, which is associated with the major excitatory neurotransmitter released in synapse of the pain-transmitting afferent neurons. Combining the pain-deregulated miRNAs (miR-145-5p and miR-200b-3p) resulted in a positive correlation with horse grimace scores in acute laminitic horses. A further study, with a small sample size, reported a decrease in levels of miR-126-5p, miR-135a-5p, miR-451 and miR-486 in cartilage and an increase in levels of miR-1249 and miR-197 in bone from three 10 month old foals with osteochondrosis compared to three control foals [19].

In terms of peptide biomarker candidate (Appendix A) studies for fracture/injury risk factors, three studies [42,43,45] reported a decreased serum level of carboxy-terminal telopeptide fragments of type II collagen (CTX-II) in in 2–3 year old Thoroughbreds following carpal or fetlock joint injuries. However, arthroscopic scores were not correlated with synovial fluid or serum CTX-II in one of these studies [45]. Furthermore, as cohort studies [42,43] the results could be subject to selection bias as a result of horses being lost to follow-up. A separate paper found the two biomarkers CTX-I and bone alkaline phosphatase were not accurate biomarkers for bone fragility syndrome represented by scapula stress fractures [44]. also reported that there was no significant difference between injury and control at the baseline or entry time point. In a longitudinal study, the greatest change of four biomarkers occurred 4–6 month prior to injury: a decrease of articular cartilage biomarkers, glycosaminoglycan (GAGs) and aggrecan chondroitin sulfate 846 epitope (CS486), and an increase in bone biomarkers, CTX-I and osteocalcin [40]. In one recent study [43], decreased osteocalcin and CTX-I, representing altered bone turnover, were specific for injured horses in a population of Polish racehorses in contrast to findings in a North American population.

### 3.5. microRNAs and Their Targets in Response to Exercise and Mechanical Loading in Horses and Humans

A review of miRNAs and their targets in response to exercise and mechanical loading in horses and humans identified 11 publications including 5 horse miRNA profiling studies [50,53,55,56,57], 1 human miRNA profiling study [49], 2 human miRNA candidate studies [52,54], 2 horse mRNA profiling studies [51,59] and 1 horse mRNA candidate study [58] (Appendix A). Overall 10 of the 11 studies were quasi-experimental studies, e.g., before and after the intervention (exercise), where miRNAs were measured at two points in time. These were not randomised, and therefore, interpolation should be carried out cautiously to similar populations. The following results should be interpreted with caution, as many of the studies only scored as low on the Grade/vetGRADE pyramid.

The three horse miRNA studies [53,56,57] and one human miRNA study [52] reported an increase in miR-21-5p levels after exercise; although, a further human study [49] reported stable expression of miR-21-5p, even after exercise. These horse studies also reported an increase in levels of members of the let-7 family in response to exercise [53,56,57]. A decrease in levels of miR-16 following an 160 km endurance competition was reported in two of these horse studies [56,57], and a gradual decrease in miR-16 levels followed by explosive strength training was also reported in one human study [52]. Analysis of the predicted genes targeted by miRNAs increased upon exercise (e.g., miR-1, miR-133 and miR-206) [50,54], suggesting the involvement of the muscle remodelling pathway (IGF1R, EGFR, PURB, TAGLN, TMOD2, LASP1 and SGCD) [50]. Two mRNA profiling studies in Arabian horses [58,59] reported that race training or competing in flat races induced an osteoclast differentiation pathway involving CTSK, IL6ST, NFATc1, CLEC5A and VAV3.

### 3.6. Genetic Association Studies Related to Stress Fractures in Horses and Young Adults (e.g., Athletes/Military Recruits) Equivalent to Equine Stress Fracture

Reviewing genetic association studies related to stress fractures in young adults (18–35 years) identified 5 human publications and 1 horse paper (Appendix A). All the identified studies were case-control studies and as such may be subject to selection bias. Four genetic studies [63,65,72,73] focused on logical candidate genes involved in bone formation and bone-associated disorders and reported that vitamin D receptor (VDR) SNPs are correlated to stress fractures; homozygotes of the rare allele of VDR SNPs are involved in the multiple stress [64], C–A haplotypes in VDR are involved in a 3.0-fold higher risk of stress fracture [63] and the F and B alleles FokI in VDR are associated with a 2.7-fold and a 2.0-fold increase in risk of stress fractures in military personnel [61]. One genome-wide association study [62] reported several different novel variants of genes including G protein-coupled receptor kinase 4 (GRK4), Nebulin (NEB), solute carrier family 6 member 18 (SLC6A18), leucine rich repeat containing 55 (LRRC55), sialic acid-binding Ig-like lectin 12 (SIGLEC12) and extracellular leucine rich repeat and fibronectin type III domain containing 2 (ELFN2) were significantly different in synovial fluid(SF) cases of Israeli soldiers compared with controls. The only horse study, by Blott et al. (2014) [60], that looked at risk of fracture in Thoroughbred racehorses (269 fractures were used and 253 controls) reported that loci in chromosome 1 and 18 were significantly associated with fracture.

### 3.7. microRNAs Related to Osteoporotic Fragility Fractures in Humans

Reviewing miRNAs related to osteoporotic fragility fractures in humans identified 11 publications, including 7 miRNA candidate studies [32,66,67,68,69,70,71] and 4 miRNA profiling studies [16,17,18,33] (Appendix A). The osteoporotic fragility fracture studies do not consistently identify differentially expressed (DE) miRNAs, for example, miR-21-5p was reported to be upregulated in three papers [17,69,70], unchanged in one paper after correction for age [68], downregulated in a further paper [67] and not measured or not significantly different in the remaining papers [18,32,33,71]. A further miRNA, miR-19b, that regulates the osteogenic transcription factor Runx2 was differentially expressed in four of the nine osteoporotic fracture miRNA studies, upregulated in three papers [32,67,71], but downregulated in one paper [18]. Mandourah et al. (2018) [16] also reported that, in combination, miR-122-3p and miR-4516 were significantly decreased in osteoporotic groups compared to osteopaenic groups and controls, but individually were of less value in diagnosing osteoporosis (AUC = 0.666 and AUC = 0.727). Feichtinger et al. (2018) reported three downregulated differentially expressed miRNAs, miR-29b-3p, miR-324-3p and miR-550a-3p, that were significantly associated to dynamic histomorphometric parameters of bone formation.

Profiling studies frequently utilise a panel of selected potential differentially expressed miRNAs as biomarkers in order to increase the sensitivity and specificity to detect bone fragility fractures. One of the earlier validation studies [17] that used miRNA PCR arrays identified miR-21, miR23a, miR-24, miR100 and miR-125 as a panel of upregulated miRNAs in both serum and bone of osteoporotic patients (20 osteoporotic patients vs. 20 non-osteoporotic controls). A larger, more recent validation study looking at osteoporotic patients with and without fracture found no association between 32 miRNAs and bone turnover markers, bone mineral density and high-resolution peripheral quantitative computed tomography. Furthermore, there was no association between baseline serum miRNA expression and prospective incident fracture in 108 women compared to 469 women without fracture [68]. Zarecki et al. [69] found seven upregulated miRNAs (miR-375, miR-532-3p, miR-19b-3p, miR-152-3p, miR-23a-3p, miR-335-5p and miR-21-5p) in patients with vertebral fractures [69]. A further validation study did find miR-29b-3p, miR-324-3p and miR-550a-3p to be correlated with bone histomorphometry and microarchitecture [66], and this work was used as a basis for a panel of 19 miRNAs that predicted more fracture events than the fracture risk assessment tool (FRAX) index [71] in a small cohort of women. The candidate and validation studies described in this section did not show consistency in the miRNA panels to differentiate between osteoporotic women with and without fracture. Direct study comparisons are difficult due to differences in sample size, study population and study design.

### 3.8. miRNA Target Analysis Related to Osteoporotic Fragility Fracture in Humans

Of the 10 studies that evaluated osteoporotic fragility factors, 4 used miRNA target prediction programmes [16,17,18,69]. Seeliger et al. (2014) [17] identified reduced PDCD4 and c-FOS influenced by miR-21, reduced Runx2 influenced by miR-23a, miR-24 and miR-27 and reduced BMPR2 influenced by miR-100 in osteoporotic patients. Mandourah (2018) [16] found miR-122-5p and miR-4516 to be correlated with fragility fracture and use of the target prediction programmes, miWalk, miRanda, RNA22 and Targetscan, with the search term osteoporosis, which suggested that eight mRNAs, including BMP2K, FSHB, IGF1R, PTHLH, Runx2, SPARC, TSC22D3 and the vitamin D receptor, would be potential targets of both miR-122-5p and miR-4516. Because targets were predicted by bioinformatically, further experimental validation (e.g., qPCR, in vitro assay) is warranted. Two more recent papers [18,69] found differential expression of miR-19b, but, although both used similar target prediction programmes including DIANA (https://bio.tools/DIANA-microT (accessed on 28 March 2021)), they identified different potential target genes for this miRNA. Five studies attempted to correlate circulating miRNAs and bone turnover markers [18,32,68,69,70]. Two studies used cell transfection experiments [18,33] and one treated mice with an agomir [18].

## 4. Discussion

This review assessed the current evidence on, and feasibility of, using reported miRNAs and mRNA targets for potential diagnostic biomarkers to predict fractures. Very few high-quality studies exist in the equine and human literature. In total, 21 equine studies and 19 human studies were selected for this systematic review.

### 4.1. Key Findings

Objective (1): MiR-21 and miR-125b were frequently reported miRNAs affected by exercise and mechanical loading in horses and humans. Blood for circulating miRNA analysis should be collected from fasted animals before exercise and samples with evidence of haemolysis should be discarded. Objective (2): There were differentially expressed miRNA, mRNA, peptide biomarkers or genetic associations reported as statistically significant across the 40 papers. There was no consensus of miRNAs to predict musculoskeletal injury in horses. Genetic association studies were often underpowered, were subject to bias and were not sex specific. Objective (3): differentially expressed miRNAs and predicted targets were not consistent between human osteoporotic fragility fracture studies.

The two blood fractions, plasma and serum, are valuable sources for the investigation of potential biomarkers in various pathological conditions. Although they are usually considered to be equivalent, several studies addressed how the type of blood fractions (serum or plasma) influences the analysis of circulating miRNA and how their miRNA content may significantly differ [74,75]. Plasma must be carefully separated from the red blood cells or buffy coat in order to avoid cellular contamination. For serum this is not the case due to the coagulation of the sample. It is still controversial which blood fraction is the most reliable source. One study showed higher miRNA concentrations in serum samples compared to the corresponding plasma samples using a qPCR platform, and this was associated with miRNA from platelets, indicating that a coagulation process may affect the spectrum of extracellular miRNA in blood [75]. Interference with polymerase chain reactions has been reported for heparinised blood samples [76]. Furthermore, using anti-coagulant heparin could complicate transcriptomic analysis because heparin may inhibit RNA polymerase so that it causes inefficient mRNA synthesis and fluorophore labelling [77]. Therefore, ethylenediaminetetra acetic acid (EDTA) is recommended when collecting blood from horses for next generation sequencing. Another study demonstrated that plasma from rats generated more aligned reads than serum from rats according to high-throughput sequencing; however, these differences were not observed in human samples [74]. In our experience, horse blood tends to form more fibrin clots due to a prolonged clotting time and poor clot retraction, which fails to reliably obtain a high quality of serum. Caution must be taken when comparing miRNA data generated from different sample types or measurement platforms.

All of the post-exercise plasma samples and the majority of the pre-exercise plasma samples underwent haemolysis in one study [53], highlighting the need for standardised quality control and sampling times in fit racehorses. No haemolysis was reported in pre- or post-blood samples in 14 Arabians that were sampled within 30 min of completing a 130 km endurance event [56]. In contrast to the Thoroughbred study where EDTA tubes were used, PAXgene Blood RNA tubes were used in the Arabian study. One in vitro study identified miR-378 and miR-30c to be haemolysis independent and identified that miR-320 and miR 324-3p were associated with red blood cell contamination [78]. Several human studies [16,32,69] controlled for haemolysis by checking a delta Cq between miR-23a and miR-451 (>7 is an indicator of haemolysis). Faraldi et al. (2019) [79] also investigated four different normalisation strategies to identify the best method for the most reliable results. Interestingly, in this study, miR-320d was found to be the most stable reference gene for circulating miRNAs in the study of sprint training in people. These authors also showed that different normalisation strategies produced different results, hence highlighting the importance of accurately choosing the most reliable normalisation methods in every individual experimental setting. The use of such techniques is considered advantageous when assessing circulating miRNAs from fit Thoroughbred racehorses. An independent validation set of animals confirmed that miR-21-5p, miR-181-5p and miR-505-5p are candidate molecules reflecting adaptation to exercise in endurance horses [56]. A separate study of four Arabian horses, sampled pre- and post-endurance competitions (90 km) using blood collected into EDTA tubes, was not in agreement with these findings and identified miR-206, miR-208b and miR-1 as the most upregulated differentially expressed miRNAs [50]. Levels of 12 circulating miRNAs (including miR-125b and miR-122-5p) were elevated immediately after marathon running in middle-aged individuals [80]. In addition to the effects of immediate exercise, human studies have evaluated the effects of different types of exercise on circulating miRNAs. One human study evaluating circulating miRNAs in endurance and strength athletes found that miR-222, miR-21 and miR-146a were significantly elevated in the endurance athletes compared to the strength athletes [81].

There is increasing work demonstrating the influence of miR-21 on bone homeostasis, the miR-21 knockout mouse showed impaired formation of the calvarial bone and in vitro work has demonstrated a dual action of miR-21 on both osteoblasts and osteoclasts [82]. MicroRNA-21 has been shown to be a sensitive and specific marker for the early identification of breast cancer in one meta-analysis [83], colorectal cancer in another meta-analysis [9] and overexpression of miR-21 was reported in both prostate cancer and osteosarcoma studies [82]. Furthermore, miR-21 has been shown to be increased in experimental studies during fracture repair in mice [84]. One candidate study [14] and two profiling studies [17,69] highlighted increased miR-21 as part of a profile of increased miRNAs in women with osteoporotic fractures. Three further studies on osteoporotic fracture did not identify increased miR-21 [15,16,18], highlighting the need for comparable sampling and processing protocols. Increased miR-21 has also been correlated with reduced bone mineral density in osteoporotic patients independent of sex [14].

There was no common miRNA reported in stress fracture/musculoskeletal injuries in horses, possibly because of the limited number of studies. The plasma- and serum-derived miRNA profile pattern in laminitic horses differed from the control; however, their miRNA targets were related to pain-related genes in the glutamatergic pathway rather than bone-related genes [20]. Injured tendon- [21], cartilage- and bone-derived miRNA profiling [19] and their targets in horses were correlated to cartilage integrity (cell cycle, differentiation, collagen production, energy production and metabolism and extracellular matrix structure) and bone maintenance (osteoblasts and osteoclasts differentiation, energy production, vesicle transport and growth factor signalling pathways). However, because of the very limited studies in horses, extensive further study is required. Pacholewska et al. [85] has compared miRNA expression in various normal tissues (abundance of miR-486 in serum and blood, miR-144 in bone, miR-20b in bone and liver, novel-miR-174, novel-miR-27, novel-miR-634 in cartilage and miR-193 in liver) and identified tissues/breed-specific miRNAs (let-7f in 30 Warmbloods, increase of miR-122 and decrease of miR-328 miRNAs in 5 ponies).

Differentially expressed miRNAs and genes were defined on the basis of the results of statistical analysis. Only significantly differentially expressed miRNAs and their targets were highlighted in cross-comparison studies. The miRNA target was validated using a qPCR of tissues to detect levels of mRNA or combination of transcriptomic data, metabolomic profiles or measurement of metabolites [49,56,57]. In vitro cellular validation was only carried out in three papers [18,19,21].

Overall, 10 studies out of 23 miRNA studies including, 12 non-biased miRNA profiling studies and 11 targeted miRNA candidate studies, carried out the validation of the findings with an independent cohort. Of these, a non-biased miRNAs profiling study in humans by Sun et al. [18] performed a comprehensive investigation of miR-19b involvement in fracture risk by the identification of differentially expressed miRNA, and validation in an independent cohort of the effect of miR-19b mimic or inhibitor and its target on osteoblastic differentiation of human mesenchymal stem cells and mouse osteoblast, in vivo in mice.

Zarecki et al. [69] showed miR-21-5p interacts with the yes-associated protein (YAP), a transcriptional factor that negatively regulates the Hippo pathway, which has a primary role in osteoblast differentiation and also an emerging role in regulation of pre-osteoclast proliferation and differentiation of osteoclast. Zarecki et al. further showed that miR-19b influences the oestrogen signalling pathway and this was confirmed in a separate study by the administration of the synthetic mimic of miR-19b (agomiR-19b) to ovariectomised mice [18]. Whilst this specific miRNA is unlikely to be of importance in equine stress fractures, these studies highlight the proof of the principal concept for miRNA target interactions. Makitie et al. (2018) [15] reported that circulating miRNAs were significantly different in the WNT1 mutation group, which leads to severe early-onset and progressive osteoporosis. Of these differentially expressed miRNAs, miR-22-3p, miR-34a-5p and miR-31-5p are known inhibitors of WNT signalling: miR-22-3p and miR-34a-5p target WNT1 messenger RNA, and miR-31-5p is predicted to bind to WNT1 30UTR. This study provided a close insight of miRNAs and their targets and their respective relations to bone disorders.

Castanheira and their colleagues [23] revealed 22 differentially expressed synovial fluid small non-coding RNAs from the metacarpophalangeal joints of early osteoarthritis compared to the control, which may provide a potential usage of synovial fluid small non-coding RNAs as molecular biomarkers for early disease in equine osteoarthritic joints. Balaskas et al. [22] demonstrated 83 differentially expressed small non-coding RNAs between the young and old equine chondrocytes that were normal on gross appearance, highlighting the importance of age-matched controls when planning miRNA studies. One recent systematic review of tendon epigenetics [24] suggested that epigenetic mechanisms, including changes of non-coding RNA, are important in predisposing ageing tendons to injury. Sex has also been shown to influence gene expression including miRNA expression [86] and must therefore be carefully considered when planning any omics studies [87].

Blott et al. (2014) [60] reported loci in chromosome 1 and 18 were significantly associated with risk of fracture in Thoroughbred racehorses (269 fractures were used and 253 controls). Statistical significance was not reached for any of the SNP in the study by Blott et al., and those authors suggested a sample size of >1000 would be necessary. Additionally, the depth of the SNP chip used in the Blott et al. paper was modest by current standards. Two of the human studies reviewed here identified the VDR FokI as a potential SNP that may predispose individuals to fracture [63,64]. A recent systematic review and meta-analysis (performed subsequent to the inclusion dates of the current work) of gene association studies looking at fracture risk in physically active individuals suggested that many of the studies were underpowered and did not reach the Hardy–Weinberg equilibrium. Furthermore, reporting bias was described as moderate to critical in each of the 10 reviewed studies, including those reviewed here [88].

Raw data or omics data (e.g., microarray, small RNA-seq, RNA seq data, etc.) from most of recent studies in this review were deposited in online databases, allowing for free sharing and open exchange data. However, for best practice, establishing the relevant data standards including clearly stated experiment descriptions, data exchange, terminology standards and experiment execution standards must also be shared to maximise the utility of existing data.

### 4.2. Limitations

The main limitation of this systematic review is the heterogeneity of the included publications due to the lack of equine fracture omics research. Therefore, the search terms were extended to include musculoskeletal injury, gene expression studies and human studies. Stress fractures in equine athletes are thought to be equivalent to fatigue fractures or stress fractures in elite human athletes or military recruits. There has been extensive research on osteoporotic fractures in humans, particularly in the ageing population, and a large number of studies have investigated miRNAs as potential biomarkers to predict fractures. Within the 11 osteoporotic fracture publications reviewed, variations in specific miRNAs analysed and different normalisation methods prevented a meta-analysis. Only a small number of miRNAs and targets were consistently identified as affected by disease; therefore, assessment of the potential of miRNAs as biomarkers for detection of fractures requires further research. The reviewed work was not substantial enough to identify an effect of age on miRNA expression and detection in horses, as many of the studies had too small a sample size to detect the influence of age, as has been reported subsequent to the literature search for this review [22,23,24].

### 4.3. Summary of Evidence

Following review of the current literature, blood for circulating miRNA analysis was most frequently collected from fasting animals, pre-exercise. The degree of haemolysis should be closely monitored such that any samples with haemolysis can be discarded. Comparisons of recommended miRNA profiles between studies reported conflicting results, suggesting there is currently no miRNA profile that can accurately predict fracture. Direct study comparisons remain difficult due to differences in the sample size, study population and molecular analysis techniques used.

## 5. Conclusions

This review highlighted the complexities of circulating miRNA analysis and genetic association studies. The key considerations when planning a genetic association study include a power calculation, sex-specific analysis and consideration of the effects of nutrition. In order to achieve high-quality, reproducible miRNA results, researchers must strive to minimise differences caused by pre-analytical factors (fasting, exercise and diet), analytical factors (whether the platform used next generation sequencing, miRNA array or qPCR) and post-analytical factors (consideration of the most stable reference gene(s) for individual studies). Despite a robust study design, including power calculations and a large sample size of several human studies, a reliable panel of miRNAs to detect and/or predict fracture occurrence was not identified.

## Figures and Tables

**Figure 1 animals-11-00959-f001:**
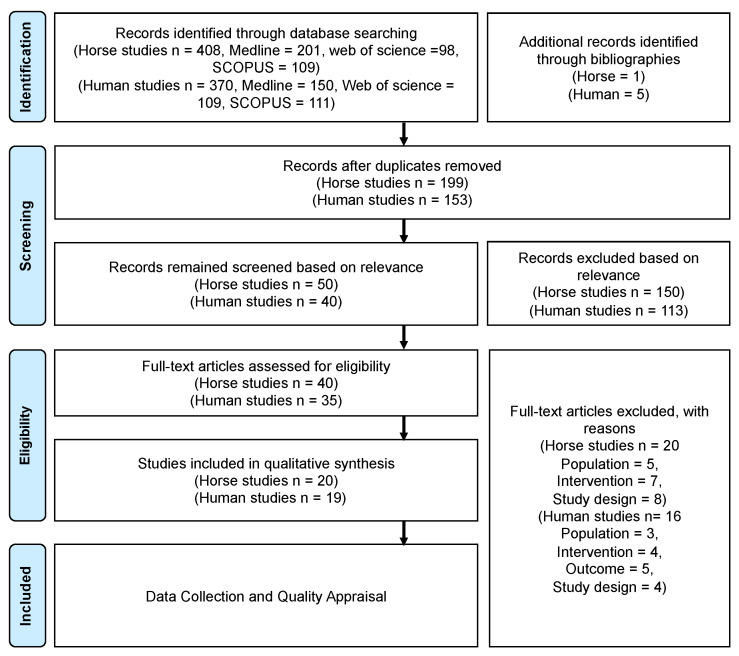
Preferred reporting items for systematic reviews and meta-analyses (PRISMA) 2009 flow diagram for the numbers of studies identified, screened, assessed for eligibility and included in this systematic review.

## Data Availability

Publicly available publications were analysed in this study. The data can be found via their citation information presented in full in the References section of the present publication.

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
