# Peer review of "Use of Omics Data in Fracture Prediction; a Scoping and Systematic Review in Horses and Humans"

_animals, 2021, doi:10.3390/ani11040959_

Round 1
Reviewer 1 Report
This manuscript reviewed the studies for fractures in human and horses, and then following interesting aspects were reported;
1) MiR-21 and miR-125b were frequently reported miRNAs affected by exercise and mechanical loading in horses and human.
2) There were differentially expressed miRNA, mRNA, peptide biomarkers or genetic associations reported as statistically significant through the systematic review using the 40 papers. However, there was no consensus miRNAs to predict musculoskeletal injury in horses. 3) Differentially expressed miRNAs and predicted targets were not consistent between human osteoporotic fragility fracture studies.
From these, the authors suggested the difficulty to predict fracture occurrence using biomarkers.
Although this reviewer considered that the current version of manuscript may be suitable for publication in the animals, the authors should be revised the following comments
The resolution of Fig. 1 is poor, when looking at the submitted manuscript.
Although all the results were described in the main text and supplemental information, is it possible to show the summarizing (between human and horse studies) as one table in the main text?
Line 377 & Line 588: The authors mentioned that Blott et al. (2014) is an only study for fracture in horses. But, there was a following article based on genetic study (https://pubmed.ncbi.nlm.nih.gov/31612520/). This is omitted based on your selection criteria?
Line 452-464: Is it possible to add comments the differences between EDTA and heparin blood collection tubes for plasma collection in miRNA study?
Is it possible to comment the relationship biomarkers and exercise intensity?
Author Response
Many thanks for the clear and constructive comments, please see our reponses below.
Reviewer 1 comments Animals stress fracture microRNA review
This manuscript reviewed the studies for fractures in human and horses, and then following interesting aspects were reported;
1) MiR-21 and miR-125b were frequently reported miRNAs affected by exercise and mechanical loading in horses and human.
2) There were differentially expressed miRNA, mRNA, peptide biomarkers or genetic associations reported as statistically significant through the systematic review using the 40 papers. However, there was no consensus miRNAs to predict musculoskeletal injury in horses. 3) Differentially expressed miRNAs and predicted targets were not consistent between human osteoporotic fragility fracture studies.
From these, the authors suggested the difficulty to predict fracture occurrence using biomarkers.
Although this reviewer considered that the current version of manuscript may be suitable for publication in the animals, the authors should be revised the following comments
The resolution of Fig. 1 is poor, when looking at the submitted manuscript.
This figure was made in power point and saved as a PDF in accordance with the journal instructions.
We have now embedded the pptx version in the template word document and hope this may improve the resolution.
Although all the results were described in the main text and supplemental information, is it possible to show the summarizing (between human and horse studies) as one table in the main text?
The summary table is 23 pages long (Supplementary Table 7) we felt this too large to include in the main text.
Line 377 & Line 588: The authors mentioned that Blott et al. (2014) is an only study for fracture in horses. But, there was a following article based on genetic study (https://pubmed.ncbi.nlm.nih.gov/31612520/). This is omitted based on your selection criteria?
When we tried to search, the following paper (https://pubmed.ncbi.nlm.nih.gov/31612520/) did not appear therefore, we did not omit intentionally. However, when we looked to evaluate and include this this paper the ethical approval/consent statement was not clearly stated in the paper preventing inclusion.
Line 452-464: Is it possible to add comments the differences between EDTA and heparin blood collection tubes for plasma collection in miRNA study?
Two sentences have been added including references at line 513 in the revised document.
Interference with polymerase chain reaction has been reported for heparinized blood samples [76]. Furthermore, using anti-coagulant heparin could complicate transcriptomic analysis since heparin may inhibit RNA polymerase so that it causes inefficient mRNA synthesis and fluorophore labelling[77]. Therefore, EDTA is recommended when collecting blood for next generation sequencing.
Is it possible to comment the relationship biomarkers and exercise intensity?
In the horse the most pronounced effects of exercise on biomarkers of synovial fluid are seen with high intensity, long term exercise (Moller and van Weeran 2017).
In terms of osteoporosis even moderate walking exercise resulted in sustained increase in urinary cross-linked N-terminal telopeptides of type I collagen from 3-12 months compared to the control group (Yamazaki et al 2004).

Reviewer 2 Report
Comments to the Authors
A brief summary
The aim of this manuscript was to provide a systematic review of scientific studies investigating molecular markers in bone fractures in horses and human, as well as other musculoskeletal disorders in horses. The main interest of this review were microRNAs, however, it concerns also gene expression, genome-wide association studies and protein markers. The review investigates the aspects of study design that may affect the outcomes of miRNA studies and formulates useful guidelines for improving quality of the future research in this subject.
Broad comments
This is a rigorously conducted systematic review highlighting significant gap in knowledge. As the authors pointed out, there are no studies investigating miRNAs as potential biomarkers of stress fractures in horses, although similar work had been undertaken in osteoarthritis, tendinopathy and laminitis. That is remarkable in the context of how serious the health and welfare consequences of stress factures are in horses and humans and justifies the place of this review within a broader literature context.
Although the methods are generally well reported and transparent, the rationale for some of the search strategies seems contrived. The introduction suggests miRNAs are the main subject of this review, then in objective 2 (Line 102) the authors mention ‘the omics data’ without explaining what that means and what methodologies will be concerned (RNAseq? Mass spectrometry? NMR?). Peptide markers, gene expression and GWAS are mentioned for the first time in paragraph 2.2. which is confusing as all the information presented prior to that focused on miRNA analysis (the abstract states that miRNAs are the only subject of the review). Inclusion of protein biomarkers in the review is surprising after the authors stated in the abstract that ‘Peptide biomarker studies have failed to consistently predict bone injury’. Protein markers also do not seem to be included in the presented search terms, unlike miRNA, RNA-seq and GWAS, and it is not clear how they ended up within the search output. The search terms also do not include exercise or mechanical loading, so it is unclear how studies concerning that aspect of the search have been identified.
In the Methods section the authors state that they broadened the search to human studies only after the search for equine studies did not produce any results. That undermines the statement made in the Introduction on how including human studies in the review is purposeful and it may inform the future equine studies. Inclusion of human studies in the objective 2nd suggests that it was a deliberate search strategy rather than result of failing to identify relevant equine studies in the first round of searches.
This review is valuable contribution to the existing knowledge base, however, I would recommend that the authors re-organise the writing to make clear what species and types of biomarkers were targeted in the search and why. That should also correspond with the search strategies described in the Methods section.
Specific comments
Line 11: Omics studies also concern metabolites (metabolomics). The more accurate and widely accepted definition of miRNA is ‘a class of small noncoding RNAs’. It may be also useful to define what authors mean by ‘genetic material’ (DNA and mRNA?).
Line 34: Correct to ‘affects’
Lines 102-105: The 3rd objective should be rephrased as it currently doesn’t make grammatical sense.
Line 166: Can the authors please specify how stress fractures in horses are similar to those in humans? Is similarity based on location, clinical presentation, radiographic image etc.? What is the evidence suggesting similar pathogenesis?
Parallels to stress fractures in human athletes should be included in the Introduction to demonstrate that expanding search to human studies was a well-justified and planned part of this review.
Line 126: What do the authors mean by ‘osseus musculoskeletal disease’? Based on search terms listed later in the manuscript it seems like any musculoskeletal disease was considered in this review.
Lines 128 and 130-131: This is the first time peptide analysis and GWAS is mentioned in the manuscript. Why have these outcomes been included in a review concerning relation between miRNA and musculoskeletal disease? Protein biomarkers do not seem to feature in the search terms (Paragraph 2.4.2.1.) – how have these studies been identified?
Line 160: Previously the authors mention that search in equine studies has been broadened to those investigating the effect of exercise/mechanical loading on miRNAs. That does not seem to be reflected in the search terms, or did I miss something here?
Line 199: The first supplementary table mentioned in manuscript text is Table 6. Please mind that according to the journal guidelines all Figures, Schemes and Tables (…) must be numbered following their number of appearance.
Line 235: Did the authors specify primary outcome 1a (and 1b, mentioned later) before?
Lines 244-255: It may not be necessary to reference Table S3 after every statement in this paragraph. It should be sufficient to include one statement at the start or end of this paragraph saying that all the supporting data can be found in that table. The same applies to the next paragraph and Table S4.
Line 300: Editing mistake, ‘target to’
Lines 457-458: What the authors mean by ‘appears much better’ here?
Lines: 488 Middle-aged, editing error.
Line 594: Did authors mean ‘show a degree of haemolysis’?
Line 631: This seems to be copy/pasted definition of the content of Acknowledgment section rather than actual acknowledgements.
Supplementary files could be better labelled/organised. The labelling of tables as presented in paragraph ‘Supplementary Materials’ (Line 610) does not agree with the labelling in the actual supplementary file e.g. Table S1 is Table 1c in SF, Table S2 is not labelled at all etc. The table description (legend) also does not seem to agree with the one presented in SF which is confusing. In case of Table 1c (S1?) it does not seem to correspond with the data presented in the table.
Due to the size of the tables it would be easier to identify them in the table number and legend was given at the top of the table, not the bottom. Table with PRISMA criteria and table presenting inclusion/exclusion scores for the considered publications are not labelled at all which makes it difficult to reference them and find in the supplementary file.
If the journal allows for submission of multiple supplementary files I would recommend separating each table into an individual file, labelled accordingly.
Author Response
Many thanks for your clear and constructive comments, please see our responses below.
We have tried to respond to your comments to make the introduction, methods and supplementary tables more in line with a review of omics. Please see our responses to your specific comments below
Specific comments
Line 11: Omics studies also concern metabolites (metabolomics). The more accurate and widely accepted definition of miRNA is ‘a class of small noncoding RNAs’. It may be also useful to define what authors mean by ‘genetic material’ (DNA and mRNA?).
Corrected such that sentence reads as follows:
Omics studies describe the study of protein, genetic material (both DNA and RNA including microRNAs (small non-coding ribonucleic acids)) and metabolites that may provide insights into the pathophysiology of disease or opportunities to monitor response to treatment when measured in bodily fluids.
Line 34: Correct to ‘affects’
Corrected thank you
Lines 102-105: The 3rd objective should be rephrased as it currently doesn’t make grammatical sense.
Corrected thank you
3) define and compare the panels of miRNA biomarkers and their miRNA targets identified in studies of osteoporotic fractures in humans to provide methodological insights for equine research
Line 166: Can the authors please specify how stress fractures in horses are similar to those in humans? Is similarity based on location, clinical presentation, radiographic image etc.? What is the evidence suggesting similar pathogenesis?
The clinical presentation and pathogenesis are similar. We have introduced this concept in the introduction:
The authors chose to consider both equines and humans as both species sustain stress fractures [25] and their bones have been shown to heal in situ without removal of the damaged domain [26].
Parallels to stress fractures in human athletes should be included in the Introduction to demonstrate that expanding search to human studies was a well-justified and planned part of this review.
Line 126: What do the authors mean by ‘osseus musculoskeletal disease’? Based on search terms listed later in the manuscript it seems like any musculoskeletal disease was considered in this review.
Corrected, thank you, the word osseous has been removed
Lines 128 and 130-131: This is the first time peptide analysis and GWAS is mentioned in the manuscript. Why have these outcomes been included in a review concerning relation between miRNA and musculoskeletal disease? Protein biomarkers do not seem to feature in the search terms (Paragraph 2.4.2.1.) – how have these studies been identified?
When these terms (miRNA* OR microRNA* OR small RNA OR RNA-seq OR GWAS) were included, we also identified many studies related to peptide biomarkers, these peptide biomarker studies have been carried out more extensively compared to recent high throughput sequencing studies.
Line 160: Previously the authors mention that search in equine studies has been broadened to those investigating the effect of exercise/mechanical loading on miRNAs. That does not seem to be reflected in the search terms, or did I miss something here?
When only miRNA term was included, horse study did not show any matched paper so we included further terms (small RNA, RNA-seq and GWA) and also broadened the term musculoskeletal and injuries. Searched results included miRNAs changes due to exercise and mechanical loading in horses not only fractures or injury related miRNAs or gene expressions. Therefore, even though we did not include the search term exercise/mechanical loading, exercise and mechanical loading related papers were identified. During the process of screening on relevance, we included these studies.
Line 199: The first supplementary table mentioned in manuscript text is Table 6. Please mind that according to the journal guidelines all Figures, Schemes and Tables (…) must be numbered following their number of appearance.
Corrected thank you. We missed the supplementary table in text. We added Line 126 and 214 (Supplementary table 1 and 2).
Line 235: Did the authors specify primary outcome 1a (and 1b, mentioned later) before?
Primary and secondary outcomes were mentioned in section 2.2 Lines 127-133, we have made the number consistent here now as 1a, 1b and 1c.
2.2. Synthesis of outcomes for the systematic review.
The three primary outcomes were (1a) changes microRNAs/ peptide analysis/ gene expression related to musculoskeletal injuries in horses (1b) microRNAs and their targets in response to exercise and mechanical loading in horses and human and (1c) Genetic association studies related to stress fractures in horses and young adults (e.g. athletes/military recruits) equivalent to equine stress fracture. The secondary outcomes were miRNA and their target analysis related to osteoporotic fragility fracture in humans.
Lines 244-255: It may not be necessary to reference Table S3 after every statement in this paragraph. It should be sufficient to include one statement at the start or end of this paragraph saying that all the supporting data can be found in that table. The same applies to the next paragraph and Table S4.
Corrected thank you
Line 300: Editing mistake, ‘target to’
Corrected thank you
Lines 457-458: What the authors mean by ‘appears much better’ here?
Corrected:
Another study demonstrated that plasma from rats generated more aligned reads than serum from rats according to high-throughput sequencing, however, these differences were not observed in human samples [74]
Lines: 488 Middle-aged, editing error.
Corrected thank you
Line 594: Did authors mean ‘show a degree of haemolysis’?
Following review of the current literature, blood for circulating miRNA analysis was most frequently collected from fasting animals, pre-exercise. The degree of haemolysis should be closely monitored such that any samples with haemolysis can be discarded.
Line 631: This seems to be copy/pasted definition of the content of Acknowledgment section rather than actual acknowledgements.
Apologies we have forgotten to include our funders! Many thanks indeed for pointing this out.
Dr Lee’s postdoctoral position investigating microRNA changes in horses with stress fractures is generously funded by the Horseracing Betting Levy Board and Dr Baker’s PhD scholarship investigating microRNA changes in horses with osteoarthritis is generously funded by The Horse Trust.
Supplementary files could be better labelled/organised. The labelling of tables as presented in paragraph ‘Supplementary Materials’ (Line 610) does not agree with the labelling in the actual supplementary file e.g. Table S1 is Table 1c in SF, Table S2 is not labelled at all etc. The table description (legend) also does not seem to agree with the one presented in SF which is confusing. In case of Table 1c (S1?) it does not seem to correspond with the data presented in the table.
The large document containing the supplementary tables has been redone such that the PRISMA-P checklist is now Supplementary table 1, Supplementary table 2 (former tables 1 and 2 amalgamated) describes the inclusion and exclusion criteria, Supplementary table 3 provides the detailed data extraction information, Supplementary table 4 provides the study and participant characteristics for the horse studies, Supplementary table 5 provides the study and participant characteristics for the human studies, Supplementary table 6 provides the Quality appraisal and risk of bias was carried out using the JBI critical appraisal tool for randomised controlled trials, Quasi-Experimental Studies, Cross Sectional Studies, Case Control studies and cohort studies and Supplementary table 7 provides a Summary of results for primary and secondary outcomes including assessment of evidence.
Due to the size of the tables it would be easier to identify them in the table number and legend was given at the top of the table, not the bottom. Table with PRISMA criteria and table presenting inclusion/exclusion scores for the considered publications are not labelled at all which makes it difficult to reference them and find in the supplementary file.
Corrected thank you
PRISMA is now table 1 mentioned in main article at Line125
Former supplementary tables 1a,b and c and 2 have been amalgamated to form Supplementary Tables 2 a-d representing the inclusion and exclusion criteria for the primary and secondary outcomes mentioned in the test at line 213.
Supplementary Table 2a describes the inclusion and exclusion criteria for primary outcome 1a changes microRNAs/ peptide analysis/ gene expression related to musculoskeletal injuries in horses, Supplementary Table 2b describes the inclusion and exclusion criteria for primary outcome 1b microRNAs and their targets in response to exercise and mechanical loading in horses and human, Supplementary Table 2c describes the inclusion and exclusion criteria for primary outcome 1c Genetic association studies related to stress fractures in horses and young adults (e.g. athletes/military recruits) equivalent to equine stress fracture and Supplementary Table 2d describes the inclusion and exclusion criteria for the secondary outcome miRNA and their target analysis related to osteoporotic fragility fracture in humans.
If the journal allows for submission of multiple supplementary files I would recommend separating each table into an individual file, labelled accordingly.
We agree we would prefer to submit as you suggest but it was not possible with the journal upload in the current format. All supplementary information must be in one document.
